# Progress in affinity ligand-functionalized bacterial magnetosome nanoparticles for bio-immunomagnetic separation of HBsAg protein

Leila Hatami Giklou Jajan[1], Seyed Nezamedin Hosseini[1], Mohsen Abolhassani[2]*, Masoud Ghorbani [3]*

1 Pasteur Institute of Iran, Dept. of Hepatitis B Vaccine Production, Research & Production Complex, Karaj, Iran, 2 Pasteur Institute of Iran, Dept. of Immunology, Hybridoma Lab, Tehran, Iran, 3 Pasteur Institute of Iran, Department of Research and Development, Production and Research Complex, Karaj, Iran

* mghorbani2000@yahoo.com (MG); mghorbani2000@gmail.com (MA)

**Data Availability Statement:** All relevant data are within the paper and its Supporting Information files.

## Abstract

Efficient Bio-immunomagnetic separation (BIMS) of recombinant hepatitis B surface antigen (rHBsAg) with high binding capacity was studied using affinity ligand immobilized bacterial magnetosome nanoparticles (Magnetospirillum gryphiswaldense strain MSR-1 bacteria) as an immunomagnetic sorbent. Our results showed immunomagnetic adsorption, acted by affinity interactions with the immobilized monoclonal antibody, offered higher antigen adsorption and desorption capacities as compared with the commercially available immunoaffinity sorbents. Four different ligand densities of the Hep-1 monoclonal antibody were examined during covalent immobilization on Pyridyl Disulfide-functionalized magnetosome nanoparticles for HBsAg immunomagnetic separation. The average of adsorption capacity was measured as 3 mg/ml in optimized immunomagnetic sorbent (1.056 mg rHBsAg/ml immunomagneticsorbent/5.5 mg of total purified protein) and 5mg/ml in immunoaffinity sorbent (0.876 mg rHBsAg/ml immunosorbent/5.5 mg total purified protein during 8 runs. Immunomagnetic sorbent demonstrated ligand leakage levels below 3 ng Mab/Ag rHBsAg during 12 consecutive cycles of immunomagnetic separation (IMS). The results suggest that an immunomagnetic sorbent with a lower ligand density (LD = 3 mg Mab/ml matrix) could be the best substitute for the immunosorbent used in affinity purification of r-HBsAg there are significant differences in the ligand density (98.59% (p-value = 0.0182)), adsorption capacity (97.051% (p-value = 0.01834)), desorption capacity (96.06% (p-value = 0.036)) and recovery (98.97% (p-value = 0.0231)). This study indicates that the immunosorbent approach reduces the cost of purification of Hep-1 protein up to 50% as compared with 5 mg Mab/ml immunoaffinity sorbent, which is currently used in large-scale production. As well, these results demonstrate that bacterial magnetosome nanoparticles (BMs) represent a promising alternative product for the economical and efficient immobilization of proteins and the immunomagnetic separation of Biomolecules, promoting innovation in downstream processing.

**Funding:** The author(s) received no specific funding for this work.

**Competing interests:** The authors have declared that no competing interests exist.

## Introduction

The downstream processes in the purification and separation of proteinaceous biopharmaceutical products often consisted of single or multiple various chromatographic and non-chromatographic steps [1]. Although immunoaffinity chromatography (IAC) is one of the most efficient and essential specific purification methods, it suffers from some limitations, such as high costs of media and buffers and losing efficiency during several elution steps aggregation and leakage of ligands [2–5]. In column chromatography, the mass transfer properties of porous materials lead to restrictions because their application necessitates long residence durations, which reduce productivity. As a result, new affinity materials with high dynamic binding capabilities and low processing time should be considered. Magnetic particles that are nonporous are intriguing possibilities for such materials. Even from unclarified broth, magnetic separation of proteins can be simply scaled up and achieved. Commercially available high-gradient magnet separation (HGMS) equipment can quickly scale up and produce GMP-compliant magnetic separation of proteins even from the unclarified broth. Many novel methods such as aqueous two-phase systems, crystallization, charged ultrafiltration membranes, precipitation, flocculation, filtration, and magnetic separation or combining some of these methods have been recently developed to overcome the significant obstacles of IAC methods [6–12]. Immunomagnetic separation (IMS) utilizing immunomagnetic anti-HBsAg -antibody conjugated Bacterial Magnetosome nanoparticles (BMs), is a well-established, simple, rapid, cost-effective, highly sensitive, specific, and high-throughput batch separation method that can be proposed as a promising alternative for IAC to purify the biomolecules in the downstream process. Different forms of magnetic nanoparticles, particularly ligand-conjugated magnetic nanoparticles, are widely used for rapid and efficient separation of different microorganisms and biomolecules such as proteins, nucleic acids, enzymes, bacteria, and viruses. Magnetic nanoparticles can be employed to separate proteins: According to preliminary research, using nano-sized magnetic particles in the nanometer range (8–15 nm) produces a huge specific surface area of 90 $m^2/g1$. In comparison to chromatographic beads, a recent study could demonstrate the competitiveness of Protein G bound on such nanoparticles. Preparation of immunomagnetic adsorbent involves attaching the biological macromolecules as an affinity ligand to the functionalized magnetic nanoparticles [10, 13–15] that have high selectivity and capacity for the desired protein adsorption. Magnetic nanoparticles as adsorbents are mainly iron oxide nanoparticles produced by physical, chemical, and biological methods that have been equipped with monoclonal antibodies (Mab) [15–17]. Biologically or green synthesized iron oxide nanoparticles are termed bacterial magnetosome nanoparticles (BMs) [15, 18–22].

Magnetosomes are intracellular organelles produced by magnetotactic bacteria that are composed of magnetite (Fe3O4) and greigite (Fe3S4). Magnetite is a single domain iron oxide nanoparticle surrounded by lipid bilayer membrane accompanied by specific soluble and transmembrane proteins [15, 21, 23–25]. Biologically or green synthesized magnetic nanoparticles are far superior to artificially synthesized magnetite nanoparticles in terms of features, such as narrow size distribution, shape control, large active surface area, high physical and chemical purity, stability, non-pyrogenicity, high magnetism, biocompatibility, and a proper density of surface functional groups, which these makes its excellent candidate materials for a variety of bio-applications [15, 18, 25–28].

Bacterial magnetosome nanoparticles and their characterization were led to consider them as of the unique bio-magnetic nano-carriers for enzymes, drugs, antibodies, and nucleic acids for detection, isolation, and separation [8, 10, 13, 18, 20, 23, 25, 26, 28–32]. For most of these applications, BMs nanoparticles must be attached to the biomolecules, which often occur through physical, biological, or chemical methods. Bio-conjugation or immobilization is a

chemical linking method to form a robust, permanent, and stable covalent attachment between biomolecules and nanoparticles, at least one of which is a biomolecule [33–37].

Immunomagnetic separation (IMS) is a versatile method with multiple applications. Optimal monoclonal antibody (Mab) coupling is essential to achieve the best efficiency in immunomagnetic separation. The following parameters, in particular, ensure that the immunomagnetic sorbents behave consistently: coupling efficiency, elution capacity, product purity, and ligand leakage. The latter has a unique function in the pharmaceutical sector, particularly in medicines that require high, repeated doses. The proportional influence of ligand density (LD) on the specific activity of immobilized ligands, efficiency in capturing/releasing antigens, and immunosorbent costs is an important aspect to consider. The interactions between antibody-antigen molecules become more challenging with high LD, and the antigen-binding efficiency suffers [10, 38–41]. The immunomagnetic matrix needs to exhibit excellent potential in the physical-chemical stability, binding capacity, and efficient recovery for target proteins. Bacterial magnetosome nanoparticles display virtually all the desirable characteristics of a matrix to immobilize biologically active molecules. The coupling efficiency and specific activity are the two most important variables used to characterize affinity ligand sorbents. The coupling efficiency (%) is calculated by dividing the number of moles of an antibody linked by the number of moles of antibody available. The specific activity is represented as a percentage of the milligrams of antigen bound per milligram of an antibody linked. The usefulness of affinity ligand sorbents will be defined by the values of these two parameters, whether from an analytical or commercial standpoint [8, 10, 14, 38–40].

This study demonstrates the applicability of magnetic beads for technical-scale HBsAg particles (that were already semi-purified from the methylotrophic yeast P. pastoris) magnetic separation and compared the process to packed-bed chromatography with immunoaffinity resin. We evaluated antibody coupling efficiency, HBsAg adsorption, desorption, recovery efficiency, and Mab leakage as its purification performance.

## Materials and methods

### Bacterial magnetosomes and its characterization

The bacterial *Magnetospirillum gryphiswaldense* strain MSR-1 (Catalogue # 6361) was purchased from Leibniz-Institute, DSMZ-German Collection of Microorganisms and Cell Cultures (Germany), cultured for 50 h at 28˚C in the medium containing 40 mM NaNO$_3$, 25 mM sodium pyruvate, 200 μM ferrous sulfate in microaerobic condition (2–5 ppm O$_2$) was optimized previously [22]. The bacteria were collected by centrifugation (10,000 g, 8 min, at 24˚C) and suspended in phosphate buffer saline (PBS, pH 7.4). The biomass was disrupted by ultrasonication through an ultrasonic cracker (180 W, 2 s work, 3 s interval, 200 repetitions, Ningbo Hi-tech, China) and extracted with NdFeB magnets (50 mm, 20 mm, 10 mm) that produced an inhomogeneous magnetic field (2.0 Tesla) (Yonjumag-China). The collected bacterial magnetosomes nanoparticles (BMs) were washed eight to ten times with PBS while agitating via low-level ultrasonication. The purified BMs nanoparticles were dispersed into 0.1 M PBS (pH 7.4). The core thickness, core surface characterization, as well as the altered properties of the core-surface construct were examined after surface modification. Transmission electron microscopy (TEM) (JEOL 7000F, USA) and dynamic light scattering in combination, were used to provide supplementary information about size, morphology, aggregation and surface thickness of BMs nanoparticles [22].

X-ray diffraction (XRD) (EQuniox, Intel-France) was used to analyze the morphology, structure, surface, and magnetic characteristics of BMs nanoparticles [42, 43]. The average size of the magnetite crystals with the XRD pattern was calculated by Scherrer's equation D = Kλ/

**Table 1. The obtained structural parameters of XRD analysis.** The average crystallite size D of the particles is calculated from the Scherrer equation: $D = K\lambda/(\beta\cos\theta)$, where K is the Debye-Scherrer constant (0.89), $\lambda$ is the X-ray wavelength (1.54 nm), $\beta$ is the peak width of half-maximum (FWHM/2 = 2.5288 Å), and $\theta$ is the Bragg diffraction angle. Breifly, $\beta \times 3.1416/180 \rightarrow 2.52887 \times 3.1416/180 \rightarrow 0.0441$; 2 Theta = 35.546 $\rightarrow \theta = 17.773$; Cos $\theta = 0.474363$; $D = K\lambda/(\beta\cos\theta) \rightarrow D = 0.94 \times 1.54 / 0.0441 \times 0.474363 \rightarrow 69.199$ nm.

| hkl | $2\theta$ (deg.) | $d_{hkl}$ or (FWHM/2)(A°) | $D_{XRD}$ (nm) | I (%) |
|---|---|---|---|---|
| 311 | 35.546 | 2.52887 | 69.19 | 100 |

($\beta \cos \theta$), where D is the average thickness in vertical direction of the crystal face, K is Scherrer constant (K = 0.94), $\lambda$ is the wavelength of X-ray, $\beta$ is the half-high width of the diffraction peak of the sample (FWHM) (FWHM is the full width at half-maximum of the diffraction peak) [44–47] (Table 1).

## Preparation of HBsAg

The rHBsAg was created by fermenting a recombinant *Pichia pastoris* (C-226) strain in saline medium supplemented with glycerol, and methanol was used to induce its expression in the fed–batch mode, as described previously [48]. Briefly, grown yeast cells were disrupted on a bed mill after adding 1 M HCl to the homogenate and centrifuged. The supernatant was adsorbed on Hyflo SuperCell and equilibrated to pH 4.0 over 2 hours with continuous stirring. rHBsAg was eluted after the washing stage, and a semi-purified material of 15–25% purity was utilized as the starting material for immunomagnetic separation and immunoaffinity chromatography [47, 49].

## Monoclonal antibody (Mab)

Monoclonal antibody against r-HBsAg was generated previously in Balb/c mice with IgG2b isotype [50]. This Mab is utilized as an Immuno ligand in the purification of rHBsAg, which is used to make a commercially available recombinant Hepatitis B virus vaccine (Heberbiovac HBK, Heber Biotech, Cuba). Protein G affinity chromatography was used to purify this monoclonal antibody from ascites, and after a buffer exchange with Sephadex G-25 Coarse (Amersham-Pharmacia Biotech, Uppsala, Sweden), the Mab was kept at 4°C in coupling buffer (0.1 M Na2CO3/ 0.1 M NaHCO3/0.5 M NaCl-pH 8.3). The affinity constant was identified by Pierre Martineau method [51]. Briefly, r-HBsAg-coated wells were allowed to interact with purified anti-rHBsAg Mab (2 μg/ml), followed by the addition of 0.39, 6.25, 25 and 100 nM of free r-HBsAg protein and then incubated for one h at room temperature. Protein concentration was measured using Lowry et al. method [52], and IgG specific concentration was measured using a direct ELISA assay. The Purity and size of purified Mab were determined by SDS-PAGE under reducing/non-reducing conditions. The binding activity against r-HBsAg protein was analyzed by ELISA and western blotting.

## Preparation of immunomagnetic adsorbent

To prepare bio-immunomagnetic adsorbent (Immuno-magnetosome), BMs nanoparticles were initially activated using a cross-linker apparatus. Briefly, 500 mg of wet BMs nanoparticles containing $NH_2$ groups in 1 ml PBS was incubated with 90 mg of Sulfo-LC-SPDP for 90 minutes at room temperature. The excess non-reacted cross-linker was removed by NdFeB magnet. The activated BMs were reduced with DTT (115 mg/ml) for 30 minutes at room temperature to to induce free sulfhydryl groups. Subsequently, the cross-linker (30 mg of Sulfo-SMCC) was added to the Mab solution (containing 15 mg/ml Mab, 1 mM PBS; pH 7.4) and

**Fig 1. Activation of BMs with Sulfo-LC-SPDP and Mab with Sulfo-SMCC cross-linkers and their immobilization.**

the mixture was incubated for two h at room temperature [53]. Excess cross-linker was removed by dialysis (12 kDa cut-off) in immobilization buffer at 4°C overnight. Finally, 15 mg of activated Mab was added to 500 mg of activated BMs, with stirring for two h at room temperature. The immobilized Mab to BMs (Mab-BMs) were collected by the NdFeB external magnetic field and stored at 4°C (Fig 1). The immunoaffinity column was prepared using CNBr-activated Sepharose 4B (GE Healthcare Bio-Sciences AB, Sweden) as described previously [48].

## Purification of r-HBsAg with immunomagnetic and immunoaffinity adsorbents

Five-milliliter of immunosorbent samples (CNBr activated Sepharose CL-4B) were packed into the PD-10 columns (Amersham-Biosciences, Uppsala, Sweden) and equilibrated with 20 mM Tris-HCl + 3 mM EDTA, pH 6.7. To prevent column clogging, five ml immunomagnetic adsorbent was added into a standard test tube and equilibrated as above. To purify the HBsAg by both adsorbents, column and vessel were loaded with an excess of a partially purified r-HBsAg preparation according to previously standardized conditions (5.5 mg r-HBsAg/15 mg Mab) in the equilibrium buffer. After washing off the unbound antigen and impurities with wash buffer (20 mM Tris-3 mM EDTA and 1 M NaCl, pH 6.7), the pure antigen was eluted from immunoaffinity and immunomagnetic adsorbents with elution buffer (containing 20 mM Tris-3 mM EDTA, 3 M KSCN, and 1 M NaCl, pH 6.7 and 100 mM glycine, pH 2.5) for 2 h at 16°C with constant mixing at 1150 rpm. The supernatant was then removed and the neutralizing buffer (1 M Tris buffer, pH 9) was added to the solids. HBsAg recovery ($mg_{HBsAg}$ $mL_{adsorbent}^{-1}$) was quantified by reading the absorbance of eluted HBsAg at 280 nm. For each matrix material, three technical replicates were prepared (three independent experiments), and from each of them, two independent samples were measured (analytical duplicates). The

stability and reproducibility of the adsorbents were characterized by measuring the amount of antibody released and variation of their adsorption and desorption capacities during 12 consecutive purification cycles.

### Efficiency of immunomagnetic adsorbent (Mab-BMs)

In immunomagnetosome adsorbent, immobilized product was verified by ELISA assay as follows. Equal amounts of Mab-BMs and activated BMs-sulfo-LC-SPDP suspension were incubated initially with semi-purified r-HBsAg followed by the incubation with goat HRP conjugated anti-mouse antibody (Sigma, Germany) at 37°C for 1 hour. The contents were washed five times with PBST 0.05% followed by addition of (100 µl) 3, 3´, 5, 5´-tetramethyl-benzidine solution. The reaction was then stopped by adding 1 M $H_2SO_4$ and the OD was read at 450 nm. The concentration of Mab attached to BMs or Sepharose was measured by Bradford assay according to the instructions of the Bradford kit of Thermo Fisher Scientific Company (MA, USA) [48]. Briefly, 50 µl of dispersed Mab-BMs matrix was added to 1.5 ml of Coomassie Plus reagent (Thermo Fisher Scientific, MA, USA) and incubated for 5 min. After exposure to the magnetic separator, absorbance of the supernatant was measured at 595 nm. The result was compared with a standard curve of bovine gamma globulin at different concentrations of 62.5, 125, 250, 500, 750, and 1000 µg/ml.

Furthermore, the iron concentration of the Mab-BMs absorbent was measured using the potassium thiocyanate method [54]. Briefly, Mab-BMs solution (1:50) was diluted with 300 µl of 6N HCl containing 1% $H_2O_2$, in which iron was dissolved and oxidized to the ferric state. Addition of 5% potassium thiocyanate led to the formation of a red complex following the reaction of thiocyanate with Fe III that was measured through absorbance at 476 nm. The results were compared with a calibration curve of BMs suspension that was prepared in the same manner with a concentration range of 20, 40, 60, 80, 100, and 120 µg/ml.

### Physicochemical properties of immunomagnetic adsorbent (Mab-BMs)

The Fourier transform infrared spectra (FTIR; Thermo Scientific Nicolet 6700) was used to confirm the adherence of Mab to the BMs nanoparticles [27].

### Homogeneity and purity of the eluted HBsAg

Homogeneity of the eluted protein was carried out by a high performance size-exclusion liquid chromatography (HPLC-SEC). After dilution in PBS/0.25 M NaCl (pH 7.0), 10 µl of samples were loaded into a TSK G3000 PW column (7.5×600 mm; particle size 10 µm) at a flow rate of 0.2 ml/min. The chromatograms were recorded and analyzed by LaChrom D-7000 HPLC system manager v.3.1. The purity of rHBsAg was determined by SDS-PAGE (12% mini-gel) under reducing conditions.

### Statistical analysis

ANOVA tests were performed with the Statgraphic program (version 5.0), with p-values less than 0.05 considered statistically significant.

## Results and discussion

### Production and characterization of the magnetosomes

An efficient purification method for purifying rHBsAg was designed and performed in this report using a bacterial magnetosome-coupled with anti-HBsAg monoclonal antibody in comparson with the standard immunoaffinity procedure. We isolated and characterized the

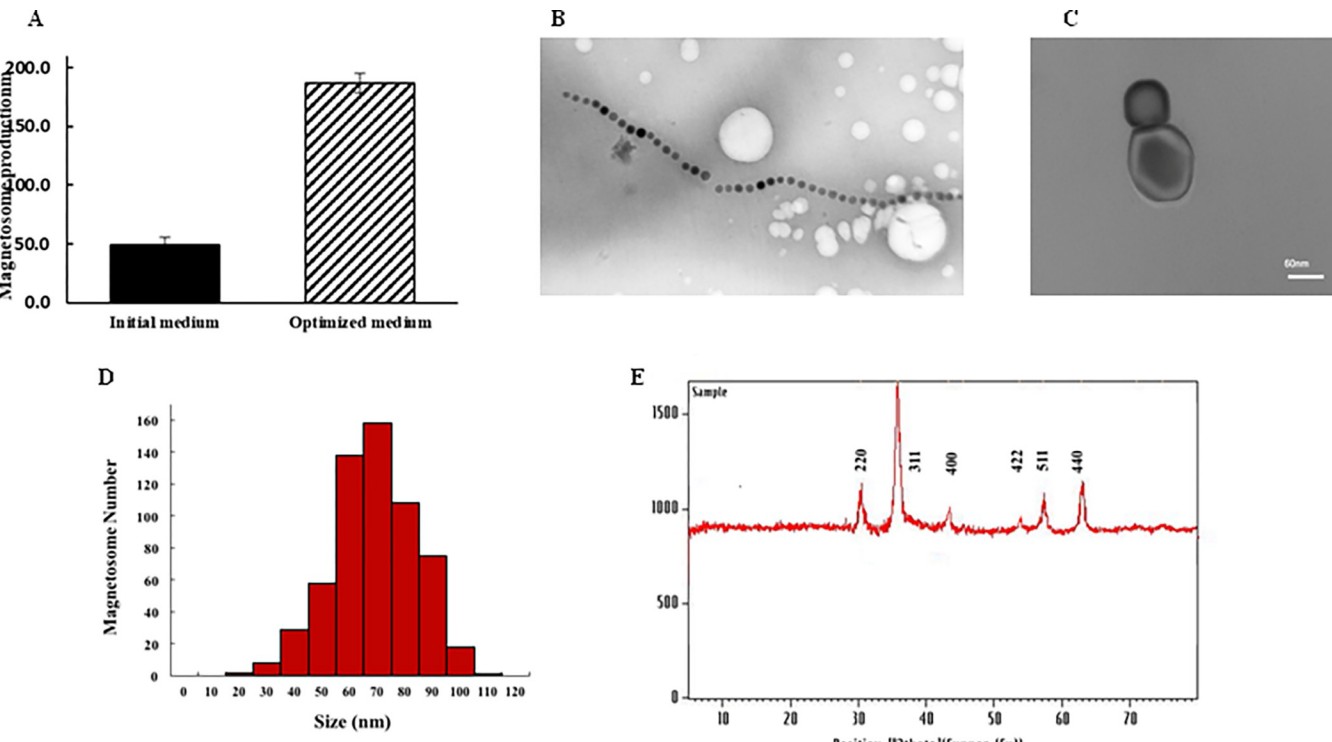

**Fig 2. Magnetosome production by *M gryphiswaldense* MSR-1 and charaterization of physicochemical properties of the BMs nanoparticles.** A) magnetosome production under initial and optimized mediums. The initial medium contains 20 mM NH4Cl, 15 mM Sodium Lactate, 100 μM Ferric citrate and microaerobic condition (5–10 ppm $O_2$) in 30˚C and pH 6.8. The optimized medium contains 40 mM NaNO₃, 25 mM Sodium pyruvate, 200 μM Ferrous sulfate and microaerobic condition (2–5 ppm $O_2$). There is a significant increase in production of magnetosome in the optimized medium (P < 0.5). B) TEM image of BMs nanoparticles chain along with the bacterial cell wall. C) TEM image of BMs nanoparticle surrounded individually by an NH2 rich phospholipid bilayer. D) BMs nanoparticle dynamic light scattering (DLS) histogram. E) X-ray pattern of BMs nanoparticles extracted from *M gryphiswaldense* MSR-1cell comprised of a core Fe3O4 magnetic molecule.

magnetosomes nanoparticles from *Magnetospirillum gryphiswaldense strain MSR-1*. The yield of extracted BMs from magnetic bacteria was around 186.87 mg/L/50h (dry weight) (Fig 2A) [22]. The TEM image of bacterium clearly shows 50–60 magnetosomes arranged in a long chain (Fig 2B). The TEM analysis of one magnetosome reveals a 60nm-80nm spherical particle with double membrane cuboctahedron (Fig 2C). These results are in agreement with the report of Araujo et al., 2015 [25]. We also measured the size distribution of magnetosome by dynamic light scattering that showed the main diameter distribution peak around 70 nm (Fig 2D). Phase purity and crystallinity of the produced BMs nanoparticles was measured by X-ray diffraction (XRD) (Fig 2E). Six characteristic diffraction peaks of a crystal structure (220, 311, 400, 422, 511, and 440) were seen that indicated the cubical spinal structure and high purity [17, 25, 34]. The average crystal size (69.19 nm) was estimated from the X-ray pattern using Scherrer's formula and line broadening measurements of the most intense peak that was near to the particle size values observed by TEM (Fig 2D). These results confirmed that the BMs nanoparticles were monocrystalline.

## Purification and characterization of the anti-rHBsAg monoclonal antibody

The advantages of the Mab CB.Hep-1 affinity ligand based separation techniques to purify rHBsAg from semi-purified starting material (15–25% purity) has been previously demonstrated in many researches [38, 48]. Murine monoclonal anti-r-HBsAg antibody (IgG2b) [50]

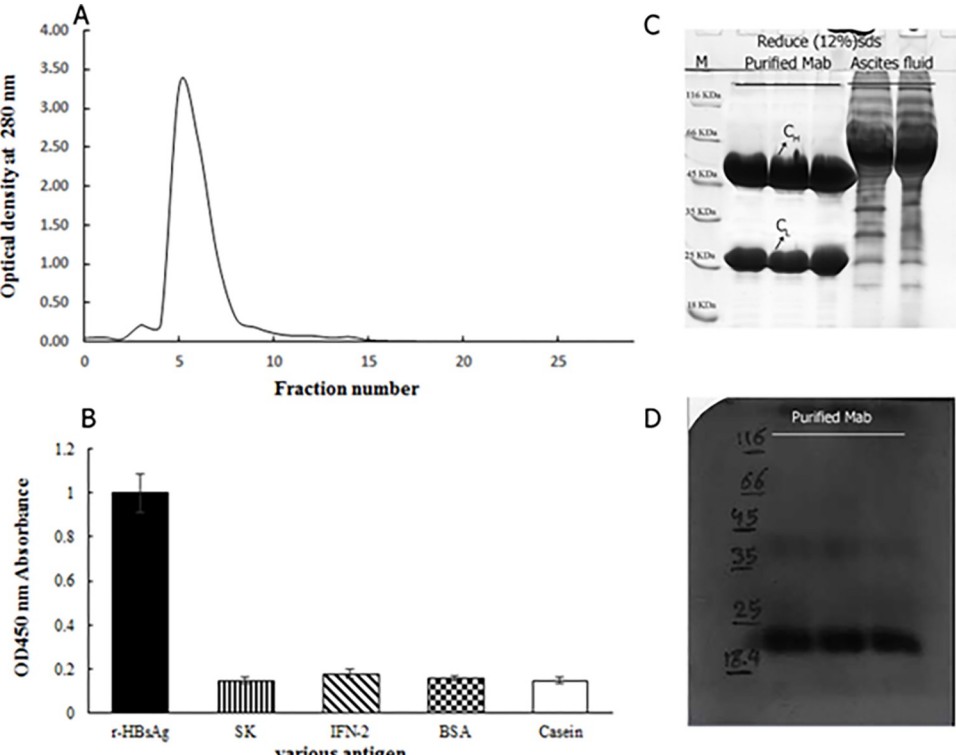

**Fig 3. Identity and specificity characterization of purified anti-r-HBsAg Mab of P1C7.** (A) The optical absorbance of purified protein. (B) ELISA distinguished specific binding to r-HBsAg protein, all used proteins including streptokinase (SK), interferon-gamma (IFN-2), and a mixture of BSA and Casein except for r-HBsAg exhibited inconsiderable absorbance. Anti-r-HBsAg Mab of P1C7 hybridoma cell line can detect r-HBsAg protein without cross-reactivity to others. (C) Coomassie Blue stained SDS PAGE (12%) of the purified Mab in reducing form: M (Ladder KD), lanes 1, 2, and 3 Purified Monoclonal antibody (Mab) or (eluted protein from protein G column), lanes 4 and 5 (ascites fluid). (D) Western blot analysis of the r-HBsAg exposed to mAb of P1C7 hybridoma cell line, M (Ladder KD), lane 1, 2, and 3 (pure r-HBsAg) after developed with purified Mab.

was purified from 86 ml ascites fluid with the yield of 9.56 μg ml$^{-1}$h$^{-1}$ (Fig 3A). The specificity of antibody was detected by ELISA as shown in Fig 3B. The affinity constant of the antibody ($7.9{*}10^{-8}$ M) was measured by Pierre Martineau method and the specificity was detected with SDS-PAGE and western blotting (Fig 3C and 3D).

## Immobilization of the Mab to magnetosomes using cross-linking

The three most common chemical approaches to immobilize biomolecules as ligand onto magnetic nanoparticles involves cross-linking, direct covalent bonding and binding to a polymeric primer [55]. In the standard affinity column, Mab was immobilized on Sepharose 4B. In this study the purified Mab was covalent directly immobilized on the surface of activated BMs support using cross-linking reagents. Multisite attachment, numerous orientations, and steric hindrance imposed by antibody crowding and antigen size are the most important variables affecting the specific activity of stochastic coupling of antibodies. Only steric hindrance affects the specific activity in covalent directed (or oriented) immobilization. By restricting the amount of protein immobilization as ligand density and the size of the antigen, the specific activity of immunosorbent produced by immobilization of affinity ligand can be improved to about 100%. The desired BMs were Ferro fluid magnetic nanoparticles that consisted of a magnetite (Fe$_3$O$_4$) core with a diameter of 70 nm. A successful immobilization Mab to the

adsorbents was determined by measuring antibody uptake during the immobilization process. In addition, the fivefold difference in ELISA absorbance between immobilized Mab to BMs and negative un-immobilized activated BMs in the IMA adsorbent evidenced a successful immobilization process (Fig 4A). For confirmation, the concentrations of the immobilized Mab and BMs on the immunomagnetic adsorbent were determined by Bradford assay and potassium thiocyanate method using dilution series of BMs suspension as a reference. The amount of 15.02 mg/ml Mab per 500 mg/ml BMs was immobilized (Fig 4B and 4C), and 25.09 mg/ml Mab to 1 g/ml packed Sepharose (Fig 4A and 4B).

The average immobilization efficiency for both BMs and Sepharose matrixes were determined as 98.59% and 94.57%, respectively (Table 2). In the present study, the extremely higher immobilization efficiency for BMs support can be related to superior features of biologically synthesized nanoparticles to synthetic magnetic nanoparticles and the use of optimum ligand density for immobilization process.

Immunomagnetic separation is an alternative method to traditional immunoaffinity adsorbent because it does not require expensive equipment, is cheaper and is stable with no significant leakage of antibody. Also, biologically synthesized magnetic nanoparticles are far superior to synthetic nanoparticles in term of narrow size distribution, shape control, large active surface area [48, 56–58]. Another advantage is non-pyrogenic and FDA approved and can be isolated easily in the magnetic field [15, 23, 25, 27, 28].

## Purification of semi-purified rHBsAg

Crude rHBsAg extract that was previously semi-purified was used for final purification using Mab-BMs and Mab-Sepharose adsorbents. The efficiency of both adsorbents regarding antigen purification was evaluated in 12 consecutive cycles and the release of antibody was measured (Table 3).

The capacity of antigen purification by Mab-BMs was higher than the regular affinity column that is mainly depends on the amount of oriented immobilized antibody, cross-linker and buffer composition used for equilibration and elution steps. It has reported that the use of heterobifunctional cross-linkers in immunomagnetosome not only provides the high specificity for the Fc portion of Mab without interrupting its antigen-binding ability, but also allows the optimal spatial orientation of the immobilized Mab because of the spacer arms [59–62]. Using the Mab-BMs adsorbent, approximately 12.67 mg antigen protein was purified during 12 cycles of purifications, whereas, using the regular affinity column, only 9.366 mg antigen was obtained. The results suggest that the immunomagnetic separation is a simple, sensitive and reproducible method for the magnetic separation of the hepatitis B protein during in the production process of the recombinant hepatitis B vaccine. The procedure has been approved by the Pasteur Institute authorities and makes it possible to comply with the requirements of the World Health Organization regarding the production of recombinant products.

## Characterization of the eluted r-HBsAg

The eluted rHBsAg obtained from both methods were first analyzed by SDS-PAGE and then confirmed by western blot analysis (Fig 5A and 5B) [63]. The purity of eluted antigen by immunomagnetosome was 98%, and with regular affinity column was 90% (Fig 5C and 5D). The SEC-HPLC profiles of the eluted rHBsAg showed three major peaks. In chromatogram (Fig 5C), the first peak appears at the retention time of 22.567 min and the third peak occurs at 43.694 min indicating aggregation and formation of the monomeric rHBsAg. The main peak with the retention time at 27.98 min represents the rHBsAg virus-like particles (VLPs) that are sharp due to the homogeneity of the particle assembly.

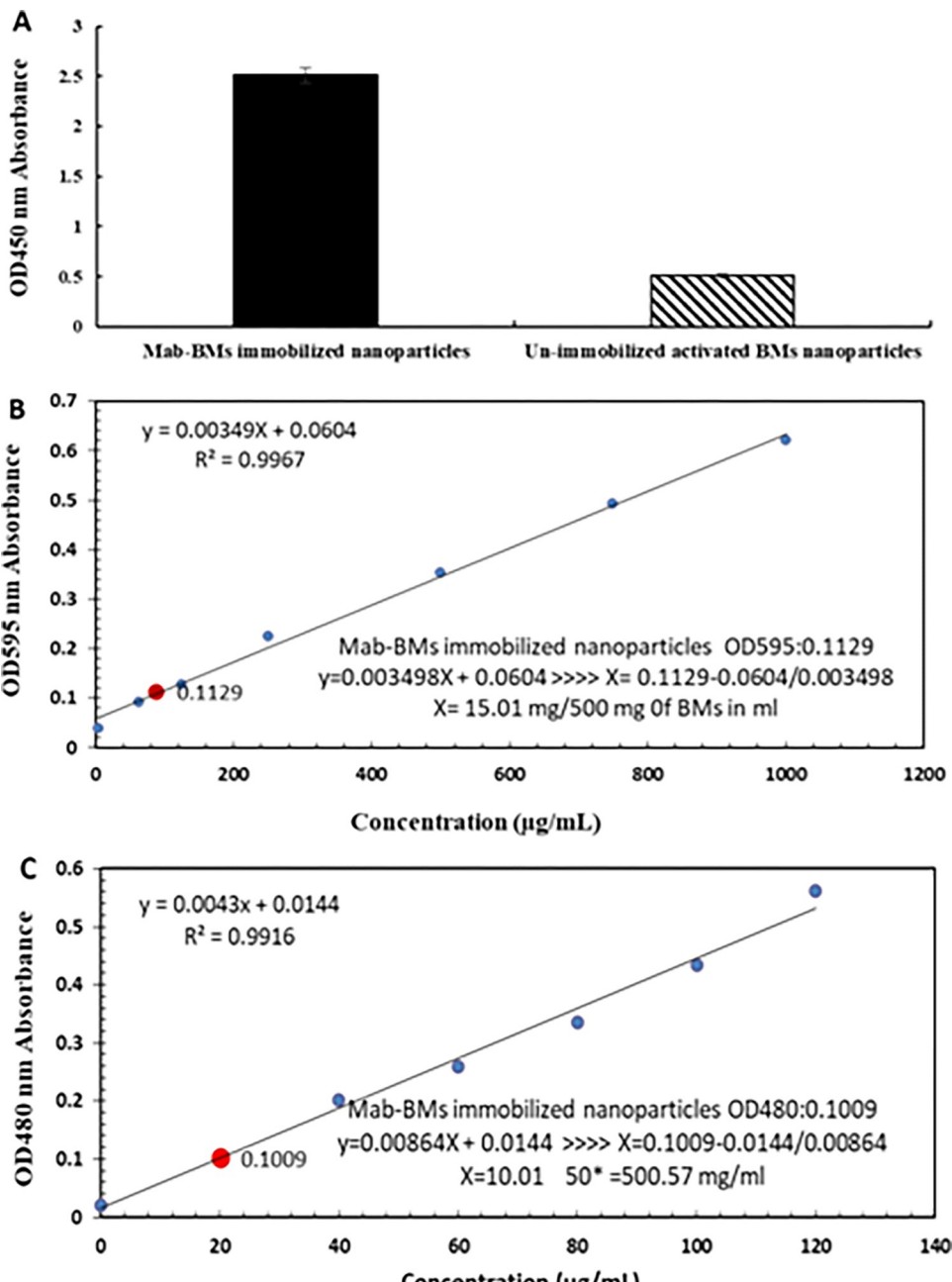

**Fig 4. Optical absorption results of ELISA test to confirm the monoclonal stabilization of the target antibody on the nanoparticle.** A) The optical absorption of the final product of the stabilized antibody on the magnetosome is 5 times higher than the activated magnetosome nanoparticles without antibody stabilization (negative control sample). B) Bradford assay showed an antibody concentration of 15.01 mg/500 mg BMs in ml. (C) BMs nanoparticles concentration of 500.57 mg/ml was obtained from potassium thiocyanate method. Strong confirmation has been made on the accuracy and performance of the stabilization process. In both experiments, results are based on a repetition as duplicate (± SD). BGG: Bovine gamma-globulin; BMs: Bacterial magnetosome; Mab: monoclonal antibody.

According to these chromatograms the purity of rHBsAg VLPs was 99.78% for immuno-magnetic and 94.36% for immunoaffinity methods (significant, p-value = 0.0006). These results also show higher recovery of rHBsAg by immunomagnetic with more effective release of the VLP form than affinity method. (Table 4).

**Table 2. Result of antibody immobilization efficiency to BMs nanoparticles as an immunomagnetic adsorbent.**

| Molecule | Amount Mab (mg) | Mab coupling (mg) | Immobilization efficiency (%) | Ligand density (mg/ml) |
|---|---|---|---|---|
| Activated Bacterial magnetosome nanoparticles (BMs) as adsorbent Average | 5 | 04.547 | 90.94 | 1 |
| | 10 | 09.302 | 93.02 | 2 |
| | 15 | 14.781 | 98.59 | 3 |
| | 20 | 19.002 | 95.01 | 4 |
| | 25 | 23.617 | 94.47 | 5 |

A volume of 5 ml was applied in each case.

## Physicochemical characteristics of immunomagnetic adsorbent

X-ray diffraction method has been used to analyze the morphology, structure, surface, and magnetic characteristics of BMs [27, 54]. This method provides information about the crystalline phase of BMs ($Fe_3O_4$) as well as the average nanoparticle diameter. Infrared analysis shows that upon activation and immobilization of BMs to Mab, the adsorption band at 3440 $cm^{-1}$ and 588 $cm^{-1}$ confirmed the existence of amino group on BMs surface and Fe–O stretching vibration respectively (Fig 6A). Presence of amino groups in BMs FTIR pattern showed that the $NH_2$ rich phospholipid membrane was preserved [43, 54]. When BMs particles were functionalizing with Sulfo-LC-SPDP as cross-linker, the FTIR spectra showed two broad bands at 3470, and 1633 $cm^{-1}$ refered to the NH stretching vibration and $NH_2$ bending state of free amine groups. As well, the FTIR spectra showed a new band of fixed propyl groups at 2923 and 2854 $cm^{-1}$ that show represented the asymmetric stretching absorption of–CH3 group that appeared at 2932 and 2862 $cm^{-1}$ in Sulfo-LC-SPDP cross-linker. These peaks demonstrate the successful cross-linker-BMs attachment.

**Table 3. Results of the evaluation of the HBsAg immunoaffinity and immunomagnetic separations experiments.**

| Runs | Dynamic capacity average (µg/200 µl) | | Adsorption capacity (%) | | Desorption capacity (%) | | Average of IgG released (ng IgG/µg Ag$^{-1}$) | |
|---|---|---|---|---|---|---|---|---|
| | IMA | IAC | IMA | IAC | IMA | IAC | IMA | IAC |
| 1 | 196.84 | 213.01 | 97.24 | 81.24 | 99.67 | 79.11 | 0.141 | 0.872 |
| 2 | 185.72 | 173.29 | 96.72 | 76.48 | 96.35 | 75.53 | 0.105 | 3.28 |
| 3 | 202.24 | 198.11 | 96.48 | 79.02 | 93.39 | 81.13 | 0.205 | 3.51 |
| 4 | 190.38 | 201.05 | 97.16 | 80.16 | 96.70 | 83.24 | 0.218 | 1.05 |
| 5 | 196.04 | 192.25 | 97.01 | 78.83 | 95.82 | 79.91 | 0.183 | 1.81 |
| 6 | 195.31 | 193.72 | 97.91 | 79.96 | 97.01 | 80.41 | 0.134 | 0.791 |
| 7 | 193.60 | 186.02 | 97.47 | 78.14 | 96.23 | 79.20 | 0.209 | 0.851 |
| 8 | 189.93 | 179.13 | 96.52 | 77.91 | 96.09 | 78.91 | 0.183 | 2.91 |
| 9 | 191.85 | 167.21 | 97.32 | 51.53 | 95.77 | 58.73 | 0.179 | 4.81 |
| 10 | 192.07 | 165.07 | 97.54 | 43.19 | 95.33 | 55.97 | 0.211 | 6.69 |
| 11 | 185.21 | 160.37 | 96.90 | 40.01 | 95.15 | 52.64 | 0.199 | 7.56 |
| 12 | 184.62 | 159.51 | 96.35 | 42.75 | 95.23 | 46.78 | 0.301 | 8.875 |
| Average | 191.984 | 182.395 | 97.05 | 67.435 | 96.06 | 70.96 | 0.189 | 3.584 |
| SD | 5.0420 | 16.8421 | 0.4569 | 16.5374 | 1.4076 | 12.7036 | 0.04759 | 2.70337 |
| P-value | 0.08413 | | 0.01834 | | 0.036 | | 0.0231 | |

*The value of every parameter was measured at least three times (mean value SD).

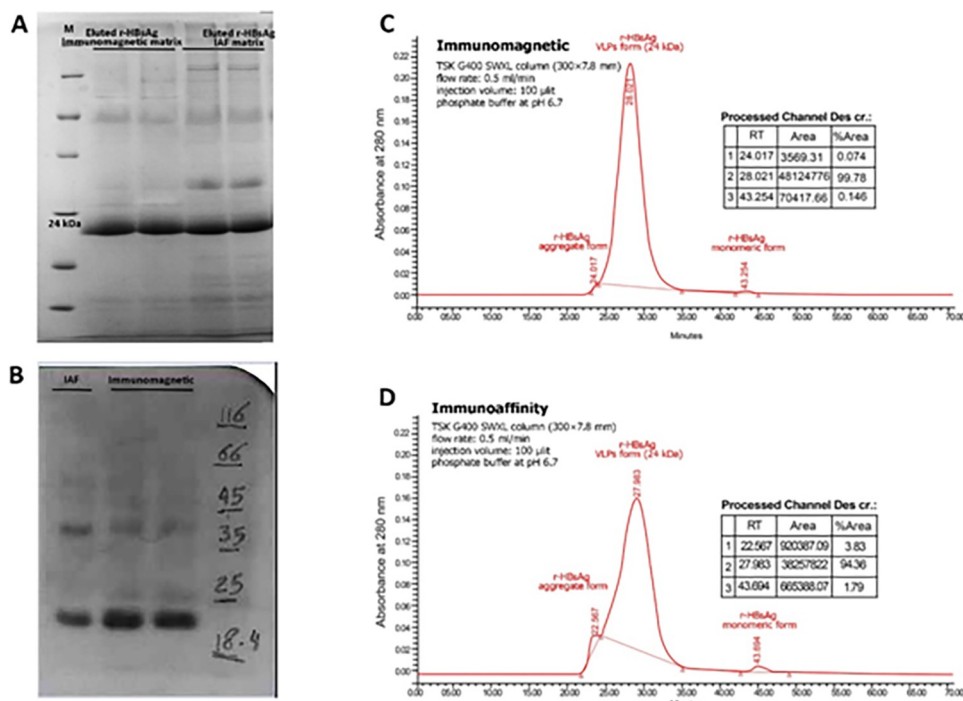

**Fig 5. Characterization of eluted rHBsAg by immunomagnetic separation (IMS) or immunoaffinity chromatography (IAC).** (A) SDS-PAGE analysis show the 24 KD band corresponding to rHBsAg purified by immunomagnetic separation method. (B) Western blot analysis show the 24 KD band corresponding to rHBsAg purified by immunomagnetic separation method. (C) and (D) Size Exclusion High-performance liquid chromatography (SEC-HPLC) analysis of eluted r-HBsAg to determine and quantify the level of VLP, monomeric and aggregated forms of eluted r-HBsAg by IMS and IAC methods, respectively.

The adsorption band at 3451 cm$^{-1}$, 1639 cm$^{-1}$ detected amino group and band at 663 cm$^{-1}$ confirmed the presence of C-OH on Mab. While the Mab functionalized with Sulfo-SMCC cross-linker via reaction of the NHS-ester group of the cross-linker with amino group in Mab Fc regain, the FTIR spectra showed new bands at 2362 and 2339 cm$^{-1}$. These bands indicated the presence of the maleimide reactive group of cross-linker that it appeared at 2360 and 2336 cm$^{-1}$ in Sulfo-SMCC cross-linker (Fig 6B).

In FTIR spectra for the Mab-BMs conjugated nanoparticles (Fig 6C), the main characteristic absorbance peaks were located at 3423, 2922, 2854, 2361, 2341, 1694, 1632, and 581 cm$^{-1}$. The bands, 3470 and 1633 cm$^{-1}$, refer to the NH and NH$_2$ groups in both activated BMs and Mab. Whereas the bands 2922 and 2854 cm$^{-1}$ show–CH$_3$ groups that are related to functionalized BMs, and the band 581 cm$^{-1}$ from BMs appeared in the final Mab-BMs conjugated nanoparticles. The broad peaks; 2922, 2854, 2361, 2341 show proper and successful attachment of BMs to Mab.

**Table 4. ANOVA results of the variable effects on the response recovery of rHBsAg by immunomagnetic and immunoaffinity matrixes.**

| ANOVA | | | | | | |
|---|---|---|---|---|---|---|
| **Source of Variation** | **SS** | **df** | **MS** | **F** | **P-value** | **F crit** |
| **Rows** | 0.3693 | 2 | 0.18465 | 6.298465 | 0.137015 | 19 |
| **Columns** | 45.76082 | 1 | 45.76082 | 1560.915 | 0.00064 | 18.51282 |
| **Error** | 0.058633 | 2 | 0.029317 | | | |
| **Total** | 46.18875 | 5 | | | | |

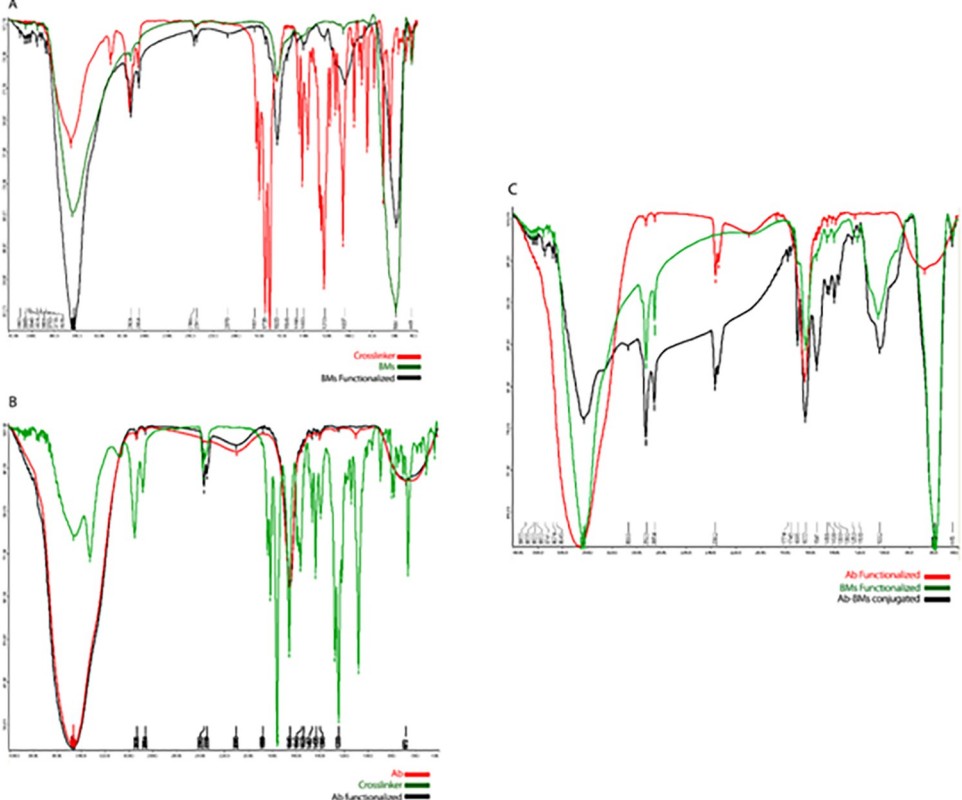

**Fig 6.** FTIR spectra of immobilization of Mab to BMs (IMA adsorbent), (A) BMs nanoparticles, Sulfo-LC-SPDP crosslinking agent and BMs functionalized (B) Mab, Sulfo SMCC crosslinking agent, Mab functionalized (C) BMs functionalized, Mab functionalized and Mab-BMs immobilized nanoparticles as a IMA adsorbent.

## Reproducibility and stability of immunomagnetic and IAF adsorbents

The immunomagnetic adsorbent was reproducible and the capturing of antigen during 12 cycles was constant and four times higher as compared with the immunoaffinity adsorbent. The immunomagnetic adsorbent was also stable over 60 days since the leakage of antibody was not significant. On the other hand, after eight cycles the efficiency of immunoaffinity was decreased due to the leakage and aggregation of antibody. However, during 60 days, the efficiency of immunomagnetic started after 12 cycles and the adsorption capacity for HBsAg was decreased slowly by time during 10, 20, 30, 45 and 60 days by 96.85%, 96.35%, 96.01, 95.92 and 95.27% respectively, as compared with the efficiency of first day (97.051%) that it is not significant (p value = 0.0056) (Table 5).

## Conclusions

In this study, we isolated magnetosomes from *Magnetospirillum gryphiswaldense* strain MSR-1 with 69.19 nm diameter and successfully developed immunomagnetic adsorbent using Mab for purification of rHBsAg. In our production facilities in Pasteur Institute of Iran, we semi-purified the r-HBsAg antigen using several expensive chromatography columns. After a few purification cycles, the column loose the efficiency due to denaturation and antibody leakage. Therefore, establishing a more economical procedure for final purification of rHBsAg appears to be necessary. The purification of antigen with immunomagnetic method was significantly higher as compared with the standard immunoaffinity method during the 12 purification

**Table 5. ANOVA results of the variable effects on the response stability of immunomagnetic adsorbent (immunomagnetic matrix).**

| Source | Sum of Squares | df | Mean Square | F-value | p-value | |
|---|---|---|---|---|---|---|
| **Model** | 1.12 | 2 | 0.5609 | 10.14 | 0.0119 | significant |
| A-time | 0.9853 | 1 | 0.9853 | 17.82 | 0.0056 | |
| $A^2$ | 0.1520 | 1 | 0.1520 | 2.75 | 0.1484 | |
| **Residual** | 0.3318 | 6 | 0.0553 | | | |
| Lack of Fit | 0.0761 | 3 | 0.0254 | 0.2977 | 0.8269 | not significant |
| Pure Error | 0.2557 | 3 | 0.0852 | | | |
| **Cor Total** | 1.45 | 8 | | | | |

R1- stability of immunomagnetic adsorbents

cycles (12.67 mg vs. 9.366 mg r-HBsAg). This method was stable, the process was reproducible and without any leakage of antibody. Immunomagnetic method is simple, rapid, sensitive, reproducible and low cost with high efficiency that could be applied as an alternative method for purification of rHBsAg from semi purified yeast extract for recombinant vaccine development industry.

## Supporting information

**S1 Video. Demonstration of the response of magnetospirillium gryphiswaldense to the magnetic field.** Magnetospirillum produce tiny magnets called magnetosomes, a type of special organelle found in bacteria's cytoplasm, which helps the bacteria to orient them in accordance with Earth's magnetic field or any other magnetic fields.
(MP4)

**S1 Raw images.**
(PDF)

## Acknowledgments

This work was performed at the production and research facility of Pasteur Institute of Iran. We would like to thank Dr. Behrouz Vaziri, Dr. Behzad Ghareyazie and Dr. Leila Ma'mani for their scientific and technical support.

## Author Contributions

**Conceptualization:** Masoud Ghorbani.

**Data curation:** Mohsen Abolhassani, Masoud Ghorbani.

**Formal analysis:** Leila Hatami Giklou Jajan, Seyed Nezamedin Hosseini.

**Investigation:** Mohsen Abolhassani.

**Methodology:** Seyed Nezamedin Hosseini, Masoud Ghorbani.

**Project administration:** Leila Hatami Giklou Jajan.

**Supervision:** Leila Hatami Giklou Jajan, Seyed Nezamedin Hosseini.

**Writing – original draft:** Leila Hatami Giklou Jajan.

**Writing – review & editing:** Masoud Ghorbani.

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
