## [Decision Letter · Decision Letter 0]

2 Feb 2022

PONE-D-21-40019Dear editorial board of Plos One,

Progress in Affinity Ligand-Functionalized Bacterial Magnetosome Nanoparticles for

Bio-immunomagnetic Separation of HBsAg proteinPLOS ONE

Dear Dr. Ghorbani,

Thank you for submitting your manuscript to PLOS ONE. I am sorry for taking so long to return this manuscript to you. I have found it very difficult to recruit reviewers over the Christmas / New Year period and rather than delaying further waiting for reviewers, I have decided to proceed on the basis of the one external reviewer I have received and my own review of the manuscript (see below). After careful consideration, we feel that it has merit but does not fully meet PLOS ONE’s publication criteria as it currently stands. Therefore, we invite you to submit a revised version of the manuscript that addresses the points raised during the review process.

If you would like to submit a revised manuscript, please do so by Mar 19 2022 11:59PM. If you will need more time than this to complete your revisions, please reply to this message or contact the journal office at plosone@plos.org. Please include the following items when submitting your revised manuscript:A rebuttal letter that responds to each point raised by the academic editor and reviewer(s). You should upload this letter as a separate file labeled 'Response to Reviewers'.A marked-up copy of your manuscript that highlights changes made to the original version. You should upload this as a separate file labeled 'Revised Manuscript with Track Changes'.An unmarked version of your revised paper without tracked changes. You should upload this as a separate file labeled 'Manuscript'.

We look forward to receiving your revised manuscript.

Kind regards,

Robert Chapman, Ph.D.

Academic Editor

PLOS ONE

Journal Requirements:

2. PLOS requires an ORCID iD for the corresponding author in Editorial Manager on papers submitted after December 6th, 2016. Please ensure that you have an ORCID iD and that it is validated in Editorial Manager. To do this, go to ‘Update my Information’ (in the upper left-hand corner of the main menu), and click on the Fetch/Validate link next to the ORCID field. This will take you to the ORCID site and allow you to create a new iD or authenticate a pre-existing iD in Editorial Manager. Please see the following video for instructions on linking an ORCID iD to your Editorial Manager account: https://www.youtube.com/watch?v=_xcclfuvtxQ.

6. Please include your tables as part of your main manuscript and remove the individual files. Please note that supplementary tables (should remain/ be uploaded) as separate "supporting information" files.

Additional Editor Comments:

As reviewer 1 points out, while the actual experimental results in this manuscript may not have been published elsewhere the work is very similar to previously published work. However, I think the work can be original in nature in its use of the platform for recovery of the antigen - to the best of my searching these results have not been published.

I do agree with reviewer 1 that criteria 3-5 for publication are not yet met.

1. The nanoparticles are characterised by a single TEM image, a number DLS histogram, and an XRD showing Fe3O4. At a minimum, better TEMs, and the full DLS data (including intensity distributions) should be provided.

2. Attachment of the protein is shown by Bradford assay, but this does not prove covalent attachment. The control in these studies should not be ‘unmodified NPs’ but rather modified NPs without the DTT treatment (or similar). Likewise the FTIR data does not prove covalent attachment as the peaks that supposedly relate to the new bonds are present in the unmodified NPs too (and at best would only show the presence of the crosslinker, not evidence that this is how the protein is attached) It is hard to see how the following conclusion is supported by the data: “the extremely higher immobilization efficiency for BMs support can be related to superior features of biologically synthesized nanoparticles to synthetic magnetic nanoparticles and the use of optimum ligand density for immobilization process”. It could be just evidence that the free unbound protein has not been washed off the surface properly?

3. The immunoseparation studies are not well described or characterised. The recovery is shown by OD at 280nm but how do we know that all of this comes from the recovered antigen? ELISA at a single datapoint (concentrations not given) is shown in figure 3, but this should be done against a standard curve to show the concentration of active antigen that elutes from the column to support the OD calculation of recovery. A much more robust study of the immunoseparation is needed.

4. The language is not ‘clear, correct, and unambiguous’ and this should also be improved before publication.

Reviewers' comments:

Reviewer's Responses to Questions

**Comments to the Author**

1. Is the manuscript technically sound, and do the data support the conclusions?

Reviewer #1: Partly

2. Has the statistical analysis been performed appropriately and rigorously? 

Reviewer #1: Yes

3. Have the authors made all data underlying the findings in their manuscript fully available?

Reviewer #1: Yes

4. Is the manuscript presented in an intelligible fashion and written in standard English?

Reviewer #1: No

5. Review Comments to the Author

Reviewer #1: The authors have attempted to separate Hepatitits B surface antigen using magnetosomes. To me, it seems an extension of previously published article (https://doi.org/10.1016/j.bbrc.2007.03.156) where the authors developed magneto immuno PCR for the detection of Hbs ag. Further, the chemistry behind the linkage, bonding and separation are not detailed. Additionanlly, the authors should do more trials and comparative experiments to support their claim; rapid, sensitive, cost effectiveness and rapidness of this separation technology. Furthermore, the language must be improved. I would suggest the authors to improve the quality of the work and resubmit.

6. PLOS authors have the option to publish the peer review history of their article (what does this mean?). If published, this will include your full peer review and any attached files.

Reviewer #1: No

---

## [Author Response · Author response to Decision Letter 0]

10 Mar 2022

Dear Dr. Robert Chapman, 

We appreciate you and the reviewers for your precious time in reviewing our paper and providing valuable comments. It was your valuable and insightful comments that led to possible improvements in the current version. The authors have carefully considered the comments and tried our best to address every one of them. We hope the manuscript after careful revisions meet your high standards. The authors welcome further constructive comments if any. Below we provide the point-by-point responses. All modifications in the manuscript have been highlighted in red. 

Sincerely,

Masoud Ghorbani, D.V.M., Ph.D.

Assistant Professor

Pasteur Institute of Iran

Journal Requirements:

1. Please ensure that your manuscript meets PLOS ONE's style requirements, including those for file naming. The PLOS ONE style templates can be found at: 

The manuscript was corrected according to PLOS ONE's style requirements.

2. PLOS requires an ORCID iD for the corresponding author in Editorial Manager on papers submitted after December 6th, 2016. Please ensure that you have an ORCID iD and that it is validated in Editorial Manager. To do this, go to ‘Update my Information’ (in the upper left-hand corner of the main menu), and click on the Fetch/Validate link next to the ORCID field. This will take you to the ORCID site and allow you to create a new iD or authenticate a pre-existing iD in Editorial Manager. Please see the following video for instructions on linking an ORCID iD to your Editorial Manager account: https://www.youtube.com/watch?v=_xcclfuvtxQ.

The ORCID ID of the corresponding author was placed in the Editorial Manager.

 The title was amended in the on line submission.

It was corrected as recommended as shown below

The phrase “data not shown” was removed and replace with Fig 3, C and D). 

Fig 3. Identity and specificity characterization of purified anti-r-HBsAg Mab of P1C7. (A) The optical absorbance of purified protein. (B) ELISA distinguished specific binding to r-HBsAg protein, all used proteins including streptokinase (SK), interferon-gamma (IFN-2), and a mixture of BSA and Casein except for r-HBsAg exhibited inconsiderable absorbance. Anti-r-HBsAg Mab of P1C7 hybridoma cell line can detect r-HBsAg protein without cross-reactivity to others. (C) Coomassie Blue stained SDS PAGE (12%) of the purified Mab in reducing form: M (Ladder KD), lanes 1, 2, and 3 (Purified Monoclonal antibody (Mab) or (eluted protein from protein G column), lanes 4 and 5 (ascites fluid). (D) Western blot analysis of the r-HBsAg exposed to mAb of P1C7 hybridoma cell line, M (Ladder KD), lane 1, 2, and 3 (pure r-HBsAg) after developed with purified Mab.

5. PLOS ONE now requires that authors provide the original uncropped and unadjusted images underlying all blot or gel results reported in a submission’s figures or Supporting Information files. This policy and the journal’s other requirements for blot/gel reporting and figure preparation are described in detail at https://journals.plos.org/plosone/s/figures#loc-blot-and-gel-reporting-requirements and https://journals.plos.org/plosone/s/figures#loc-preparing-figures-from-image-files. When you submit your revised manuscript, please ensure that your figures adhere fully to these guidelines and provide the original underlying images for all blot or gel data reported in your submission. See the following link for instructions on providing the original image data: https://journals.plos.org/plosone/s/figures#loc-original-images-for-blots-and-gels. In your cover letter, please note whether your blot/gel image data are in Supporting Information or posted at a public data repository, provide the repository URL if relevant, and provide specific details as to which raw blot/gel images, if any, are not available. Email us at plosone@plos.org if you have any questions.

All images were adjusted and corrected as much as possible and resubmitted. The original uncropped and unadjusted images underlying the blot and the gel were reported in the submission’s figure 3. 

6. Please include your tables as part of your main manuscript and remove the individual files. Please note that supplementary tables (should remain/ be uploaded) as separate "supporting information" files.

Tables were included in the body of the text as recommended

Additional Editor Comments:

As reviewer 1 points out, while the actual experimental results in this manuscript may not have been published elsewhere the work is very similar to previously published work. However, I think the work can be original in nature in its use of the platform for recovery of the antigen - to the best of my searching these results have not been published.

I do agree with reviewer 1 that criteria 3-5 for publication are not yet met.

We tried to correct all the issues quoted in criteria 3-5 in previous section.

1. The nanoparticles are characterised by a single TEM image, a number DLS histogram, and an XRD showing Fe3O4. At a minimum, better TEMs, and the full DLS data (including intensity distributions) should be provided.

Concerning your comments, a better TEM was submitted to replace the previous one. Regarding the full DLS the below table was also added to the text to clarify how the XRD was calculated. 

Table 1: The obtained structural parameters of XRD analysis. The average crystallite size D of the particles was calculated from the Scherrer equation: D = Kλ/(βcosθ), where K is the Debye-Scherrer constant (0.89), λ is the X-ray wavelength (1.54 nm), β is the peak width of half-maximum (FWHM/2 = 2.5288 Å), and θ is the Bragg diffraction angle. Breifly, β×3.1416/180 →2.52887 × 3.1416 /180→ 0.0441; 2 Theta= 35.546 → θ= 17.773; Cos θ = 0.474363; D=Kλ/ (β cos θ)→ D= 0.94 × 1.54 / 0.0441 × 0.474363 → 69.199 nm.

I (%) DXRD (nm) dhkl or (FWHM/2)(A⸰) 2θ (deg.) hkl

100 69.19 2.52887 35.546 311

2. Attachment of the protein is shown by Bradford assay, but this does not prove covalent attachment. The control in these studies should not be ‘unmodified NPs’ but rather modified NPs without the DTT treatment (or similar). Likewise the FTIR data does not prove covalent attachment as the peaks that supposedly relate to the new bonds are present in the unmodified NPs too (and at best would only show the presence of the crosslinker, not evidence that this is how the protein is attached) It is hard to see how the following conclusion is supported by the data: “the extremely higher immobilization efficiency for BMs support can be related to superior features of biologically synthesized nanoparticles to synthetic magnetic nanoparticles and the use of optimum ligand density for immobilization process”. It could be just evidence that the free unbound protein has not been washed off the surface properly? 

In respect to the Bradford assay and covalent attachment, I would like to emphasize that, we used cross-linker-activated nanoparticles as a standard to look for specific bonds that required us to use modified nanoparticles, and did not use unmodified nanoparticles due to non-specific bonds, because we were not interested in adsorption bonds, hydrogen bonds or van der Waals bonds. Regarding the links related to the FTR spectrum, it should be noted that in Figure 6 (C), the peaks are related to the 2362 wavelength of the monoclonal antibody. Whereas, the peaks 2854 and 2923 belong to the activated nanoparticles, both of which reappeared in the final product of conjugated antibody to nanoparticles. The wavelength of 1541 can be proof of the successful binding of nanoparticles to monoclonal antibodies from a specific site. Finally, in order to remove unwanted and non-specific bound proteins, the washing steps were performed very carefully and in several successive steps.

Regarding the last part of this comment, it should be noted that the wash step was repeated at least three times to eliminate the possibility of unbinding proteins in the conjugated final product. Also, using stoichiometric calculations, by reading the adsorption of the fractions obtained from the wash and the total amount of antibody used at the beginning of the reaction, and also by reading the adsorption of the conjugated antibody, confirms the absence of unbinding proteins in the final product.

3. The immunoseparation studies are not well described or characterised. The recovery is shown by OD at 280nm but how do we know that all of this comes from the recovered antigen? ELISA at a single datapoint (concentrations not given) is shown in figure 3, but this should be done against a standard curve to show the concentration of active antigen that elutes from the column to support the OD calculation of recovery. A much more robust study of the immunoseparation is needed.

With respect to your opinion, it should be noted that due to the use of specific antibodies as well as the identified antigen, due to the adsorption in OD 280 which is used to read the protein adsorption, the absorbed protein by antigen-specific antibodies, as well as complementary results in Figure 5 (where chromatographic and Western blot results are concerned), can be used to ensure that the protein obtained from the recovered yield is the antigen in question.

At last, it should be noted that due to budget constraints, time, and laboratory facilities, it is not possible to perform further complementary experiments, and we hope that in the near future we will be able to elaborate more experiments to obtain more satisfactory results to report.

4. The language is not ‘clear, correct, and unambiguous’ and this should also be improved before publication.

We had the language and the grammar checked by several interpreters and editors and did our best to remove the ambiguity and correct the grammar mistakes.

---

## [Editor Report · Decision Letter 1]

5 Apr 2022

Progress in Affinity Ligand-Functionalized Bacterial Magnetosome Nanoparticles for

Bio-immunomagnetic Separation of HBsAg protein

PONE-D-21-40019R1

Dear Dr. Ghorbani,

We’re pleased to inform you that your manuscript has been judged scientifically suitable for publication and will be formally accepted for publication once it meets all outstanding technical requirements.

Kind regards,

Robert Chapman, Ph.D.

Academic Editor

PLOS ONE

Additional Editor Comments (optional):

The authors have made no changes at all to this manuscript following the comments from the reviewers (one of which recommended a reject, and one of which asked for major revisions). They have also failed to address any of Reviewer 1's comments.

1) Characterisation of the nanoparticles: The authors claimed they replaced the TEM image but actually they just provided the same one inverted in the horizontal axis, with the brightness changed. Intensity distributions in the DLS were asked for and were not provided. The XRD does not help characterise the size and size distribution of the particles in solution.

2) Protein attachment: The FTIR does not prove covalent attachment, it just proves the presence of all of the components! The Bradford assay of the unmodified NPs are necessary if the authors want to claim (as they do) that the attachment of the protein is due to the SMCC linker and not due to non-specific interactions. This is not a very important point, and it would be fine if the authors tone down the conclusions drawn from this data - the antibody is clearly attached - the data just doesn't prove that its because of the SMCC linker.

3) Characterisation of the rHBsAb: The authors have pointed out that the SEC-HPLC analysis shows the recovered protein to be the rHBsAb. While data is not provided data to show that the protein is in its active form, this is perhaps not actually necessary to support the conclusions the paper draws.

4) No changes were made to address the language in the manuscript - the author's rebuttal on this point is just "we've had it checked and no changes are needed"!

For these reasons I do not think the manuscript is 'of a high technical quality', and it remains very similar to previously published studies. There was an opportunity to greatly improve the paper on revision if the authors had been prepared to. However, it seems to me on reading their rebuttal that the actual conclusions they draw are supported by the data and for this reason it does meet PLOSone's publication criteria.

---

## [Editor Report · Acceptance letter]

15 Jun 2022

PONE-D-21-40019R1 

Progress in Affinity Ligand-Functionalized Bacterial Magnetosome Nanoparticles for Bio-immunomagnetic Separation of HBsAg protein 

Dear Dr. Ghorbani:

I'm pleased to inform you that your manuscript has been deemed suitable for publication in PLOS ONE. Congratulations! Your manuscript is now with our production department. 

Kind regards, 

on behalf of

Dr. Robert Chapman 

Academic Editor

PLOS ONE